# Single-pixel p-graded-n junction spectrometers

Jingyi Wang [1,2,5], Beibei Pan [1,2,5], Zi Wang [1,3,4,5], Jiakai Zhang [1], Zhiqi Zhou[1,2], Lu Yao[1,2], Yanan Wu [1], Wuwei Ren [1], Jianyu Wang[1,3,4], Haiming Ji [4], Jingyi Yu [1] ✉ & Baile Chen [1,2] ✉

Ultra-compact spectrometers are becoming increasingly popular for their promising applications in biomedical analysis, environmental monitoring, and food safety. In this work, we report a single-pixel-photodetector spectrometer with a spectral range from 480 nm to 820 nm, based on the AlGaAs/GaAs p-graded-n junction with a voltage-tunable optical response. To reconstruct the optical spectrum, we propose a tailored method called Neural Spectral Fields (NSF) that leverages the unique wavelength and bias-dependent responsivity matrix. Our spectrometer achieves a high spectral wavelength accuracy of up to 0.30 nm and a spectral resolution of up to 10 nm. Additionally, we demonstrate the high spectral imaging performance of the device. The compatibility of our demonstration with the standard III-V process greatly accelerates the commercialization of miniaturized spectrometers.

Optical spectrometers are broadly deployed in scientific and industrial applications, ranging from spectral detection[1] to chemical sensing and to hyperspectral imaging[2]. Traditional solutions generally require using mechanically movable components such as optical gratings or Michelson interferometers[3,4] and therefore tend to be too bulky for onsite deployment. Tremendous efforts have been focused on miniaturization, e.g., via dispersive optics[5,6], narrow band filters[7,8] or Fourier transform interferometers[9,10]. More recent solutions have resorted to photodetector arrays, where each individual detector is equipped with a tailored optical responsivity[11-17] and the overall array recovers a broad spectrum. Array-based solutions also benefit from efficient computational techniques to either interpolate between sampled spectra (e.g., with narrow band filters[8]) or to separate the overlapping ones (e.g., with multiple detectors of different optical responses[6]). However, for each detector to sense a different spectrum, it is essential to split lights to multiple paths. More splitting increases the spectral resolutions but reduces the signal and subsequently measurement sensitivity. Striking an intricate balance between resolution and sensitivity remains challenging[18].

Instead of using an array, single-pixel-photodetector spectrometers have recently emerged as a plausible alternative. By employing electrically reconfigured response functions[19-22], these devices aim to recover the full spectra using a single optical element without splitting the light, hence offering much higher sensitivity with comparable spectral resolution. The seminal work of single nanowire[22] and two-dimensional materials junction[19-21] techniques not only show promising results but also demonstrate the ultra-compact footprints for miniaturized spectrometer application.

III-V materials have matured over the past decades and have been massively deployed in high-performance optoelectronics devices including lasers[23-26] and photodetectors[27-29]. Recent advancements have led to the development of ultra-small footprint photodetectors based on III-V semiconductors[30], and focal plane arrays (FPA) of photodetectors using III-V semiconductors have demonstrated high yields[29]. These achievements underscore the considerable potential of III-V photodetectors for mass production and integration. In this research, we seek to design a III-V material-based miniature spectrometer. However, III-V material-based spectrometers cannot directly follow lateral composition gradation[22] and stark effect[19] schemes as in

[1]School of Information Science and Technology, ShanghaiTech University, Shanghai, PR China. [2]Shanghai Engineering Research Center of Energy Efficient and Custom AI IC, Shanghai, PR China. [3]Shanghai Institute of Technical Physics, Chinese Academy of Sciences, Shanghai, PR China. [4]University of Chinese Academy of Sciences, Beijing, PR China. [5]These authors contributed equally: Jingyi Wang, Beibei Pan, Zi Wang. ✉e-mail: yujingyi@shanghaitech.edu.cn; chenbl@shanghaitech.edu.cn

nanowire or two-dimensional materials, which is difficult for III-V materials as they would require lattice matching in planar epitaxial growth.

We instead present a p-graded-n junction photodetector (pGn PD) with voltage-tunable-response based on $Al_xGa_{1-x}As/GaAs$ materials as a spectrometer. Unlike the gradient material nanowires that work in a lateral fashion[22], the p-graded-n spectrometer relies on a stacking longitudinal compositionally graded epitaxial structure and can achieve a voltage-tunable-response by varying the depletion thickness of the pn junction. The design of our p-graded-n spectrometer relies on a longitudinal epitaxial structure gradient. This means that the physical lateral size of the device does not impact its functionality. Such spectrometers can be fabricated by standard III-V process with an ultra-small footprint down to μm and a broad spectral range from 480 nm to 820 nm.

The device has a high responsivity of 0.51 A/W at 775 nm, and a low dark current density of $3.47 \times 10^{-9} A/cm^2$ at room temperature, corresponding to a calculated detectivity of $1.86 \times 10^{13} cmHz^{1/2}/W$. For spectrometer applications, the device exhibits a longer cut-off wavelength as the bias voltage increases, contributed by the newly depleted region of the junction. Thus, the junction produces a unique 'voltage accumulative' responsivity matrix: higher voltages induce broader spectral response curves. Yet, overlaps of these curves cause the spectral reconstruction problem ill-posed. Traditional methods based on L1 or L2 regularizers require elaborate parameter fine-tuning to achieve a high-resolution reconstruction.

We tailor a fully automated scheme that directly extracts deep features from the measured current-voltage curves and then conducts continuous spectral reconstruction via Neural Fields (NFs). Specifically, our feature extractor is trained on a comprehensive synthetic dataset to reliably capture unique characteristics of the current-voltage curves. These features, concatenated with wavelength positional encoding, are then fed into a Multilayer Perceptron (MLP) to decode the inputs into a continuous spectral function. The self-supervised process is analogous to neural field reconstructions for radiance fields[31], heterogeneous protein structures[32], and biological tomography[33]. We further enforce the recovered spectral function to obey physical-based, spectrum-response integral constraints and achieve spectral reconstruction accuracy up to 0.30 nm and spectral resolution up to 10 nm.

## Results
### Device design
The p-graded-n structure of the device is illustrated in Fig. 1a, with all the epi-layers being lattice-matched to the GaAs substrate. The epi-layer starts with a 500 nm n+ GaAs contact layer, with a doping concentration of $2 \times 10^{18} cm^{-3}$. This layer is followed by a 2 μm grading n-doped $Al_xGa_{1-x}As$ layer, with the Al composition gradually increasing from 0 to 0.5. (Please refer to Supplementary S2 for bandgap of $Al_xGa_{1-x}As$ material.) A 50 nm InGaP hole barrier layer was inserted within the n- $Al_{0.5}Ga_{0.5}As$ layers, followed by a p-doped $Al_{0.5}Ga_{0.5}As$ layer. The GaAs p+ layer is grown as the top contact layer. Ti/Pt/Au contact metal is deposited as the contact metal layer. The details of the device fabrication can be found in the Method section.

The p-graded-n structure enables voltage-tunable-response in this photodetector. The structure comprises a gradient bandgap $Al_xGa_{1-x}As$ n-layer, with different Al composition, where the higher bandgap $Al_xGa_{1-x}As$ is closer to the p-n junction. As illustrated in the band diagram in Fig. 1c, d, the absorption of shorter wavelength light (indicated as "green" light) starts at higher Al-composition layers which are stacked above, while these layers with larger bandgap are transparent to the longer wavelength light (indicated as "yellow" light). At low reverse bias, holes generated by longer wavelength incident light absorbed in low-Al $Al_xGa_{1-x}As$ layer are blocked by the valence band barrier from the high-Al $Al_xGa_{1-x}As$ layer and InGaP layer. Thus, these

holes cannot diffuse into the depleted region and have no contribution to the photocurrent, as shown in Fig. 1c. When the reverse bias increases, the low Al composition $Al_xGa_{1-x}As$ n region is further depleted, allowing for the photogenerated minority carriers by the longer wavelength incident light to contribute to the photocurrent, as indicated in Fig. 1d. In other words, this device architecture enables voltage-tunable-spectrum responsivity, which encodes the spectral signature of the incident light in the photocurrent-voltage characteristics. Furthermore, the p-graded-n structure design has a significant advantage in keeping the larger bandgap material near the junction, where the tunneling current is minimized under high reverse bias. This advantage is essential for photodetectors operating in longer wavelength ranges, such as the short-wavelength infrared (SWIR) or mid-wavelength infrared (MWIR) regions, which are susceptible to high tunneling currents. (Please refer to Supplementary S1 for more details of the working mechanism and advantages of the p-graded-n junction spectrometer.)

### Electrical characteristics and optical characteristics
Figure 1e presents the device's dark current and photo current density curve, revealing an extremely low dark current density of $3.47 \times 10^{-9} A/cm^2$ under a −10 V bias voltage at room temperature. As shown in Fig. 1f, g, our device exhibits responsivity across the operational wavelength range from 480 nm to 820 nm. (Please refer to Supplementary S9 for details of response function characterization.) It has a peak responsivity of 0.51 A/W at 775 nm, corresponding to the external quantum efficiency of 81%. (The detailed optical characteristics of the device can be found in Section S4 and S6 of Supplementary material.) The corresponding thermal-noise-limited detectivity[34] of the detector is $1.86 \times 10^{13} cmHz^{1/2}/W$. Due to the mature III-V technology, the photodetector performance is considerably better than the previously reported single-pixel-photodetector spectrometers[21,22]. (Comparison with recent works can be found in Section S7 in Supplementary material.)

For spectrometer applications, our device has an electrically reconfigurable response. The device exhibits a longer cutoff wavelength as the reverse bias increases, where the longer wavelength response is contributed by the newly-depleted region of lower bandgap AlGaAs materials. We encoded the variation of responsivity versus the wavelength and reverse bias into a spectral response matrix for spectrum reconstruction.

### Spectrum reconstruction
Figure 2 demonstrates the pipeline of our spectrometer to measure the unknown spectrum. By varying the bias voltage, the spectrometer integrates the incident spectrum with corresponding spectral responsivity and results in a photocurrent curve. The photocurrent measurement at a bias voltage $V$ can be expressed as the integration equation by unknown spectrum $\boldsymbol{P}(\lambda)$ and pre-calibrated spectral responsivity $\boldsymbol{R}(\lambda, V)$:

$$\boldsymbol{I}(V) = \int_{\lambda_{min}}^{\lambda_{max}} \boldsymbol{R}(\lambda,V)\boldsymbol{P}(\lambda)d\lambda \qquad (1)$$

where $\boldsymbol{I}(V)$ is the measured photocurrent curve by our spectrometer and the integration range ($\lambda_{min}, \lambda_{max}$) denote the minimum and maximum functional wavelengths of the spectrometer, respectively. (Please refer to Supplementary S8 for the detailed practical discrete spectrometer model.)

We observe that the voltage accumulative spectral responsivity produces some unique features in the measured current. And different input spectra will cause different features. Such features can play an important role in spectrum reconstruction, but they are hard to analyze by traditional spectral reconstruction algorithms. Thus, we adopt a neural feature extractor to utilize these features in measured current.

We create a synthetic dataset to provide an amount of paired data of photocurrent and optical spectrum. (Please refer to Supplementary S10 for construction of the synthetic dataset.) Then we let the feature extractor learn deep priors of photocurrent on the synthetic dataset. In Fig. 2, we also visualize the distribution of three types of spectrum currents in the feature space's principal component calculated by Principle Component Analysis (PCA).

At the core of our approach, we use Neural Spectral Fields (NSF) to decode the spectrum from the extracted features and wavelength. NSF can provide a continuous reconstruction of spectrum inputting a spectrum feature and positional encoding of wavelength without any fine-tuned efforts. To further improve the accuracy of the spectrum reconstruction, we apply a simple yet effective physics-based refinement algorithm following NSF to adjust the amplitude, width, and shift of the NSF output spectrum to obey integral constraint Eq. (1). (see

Supplementary Figs. 6 and Section S11 in Supplementary material for details of the proposed neural methods.)

## Reconstruction results

The incident light spectrum consists of single-peak and double-peak beams generated by a super-continuous laser and unknown spectra with uncertain full width at half maximum (FWHM) produced by LED lamps. The accuracy of the reconstructed spectra is evaluated against the ground truth measured with a commercial spectrometer. Our proposed approach achieves high-quality reconstruction results for a series of single-peak incident light over a broad working range of 480–820 nm, as illustrated in Fig. 3a. Next, we evaluate the performance of our single-pixel p-graded-n junction spectrometer on LED light sources. The device is measured under the LED illumination of green, yellow, and red to reconstruct the spectra, as presented in

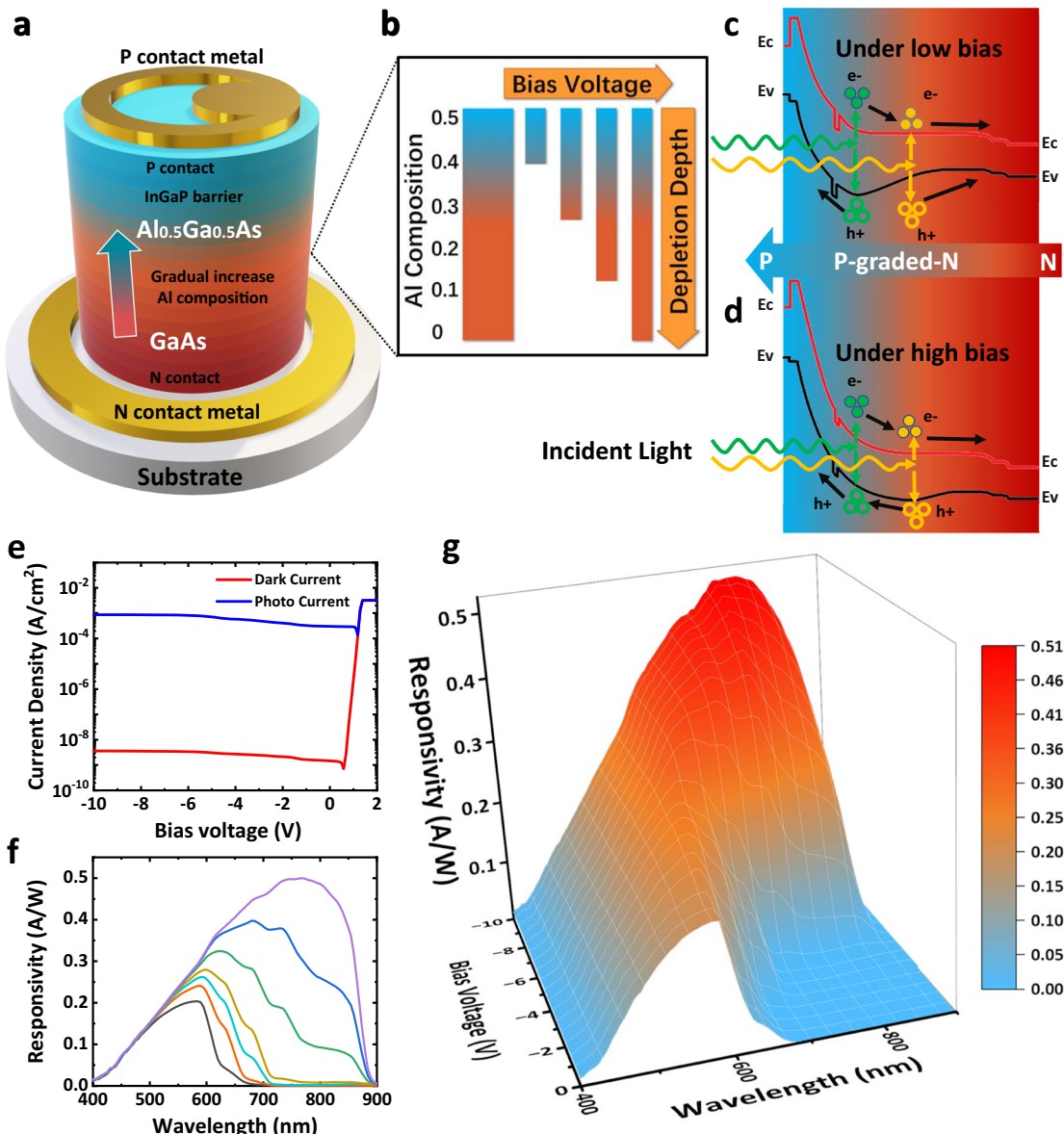

**Fig. 1 | Schematic and electric properties of the p-graded-n junction spectrometer.** **a** Schematic diagram of a p-graded-n junction spectrometer. The Al composition of the n-AlGaAs layer is graded from 0 to 0.5 along the growth direction. **b** Device working principle. As the applied reverse bias increases, the depletion region extends downwards, gradually turning on the new absorption regions with longer cut-off wavelength. **c, d** Band diagram of the p-graded-n junction spectrometer under low reverse bias (up) and high reverse bias. **e** Dark current density and photo current density of the detector. **f** Measured responsivity of the detector from 400 nm to 900 nm. Lines are the responsivity of the detector under different bias ranging from 0 V to −10 V. **g** Responsivity versus reverse bias and wavelength. The detector exhibits a longer wavelength response as the reverse bias voltage increases.

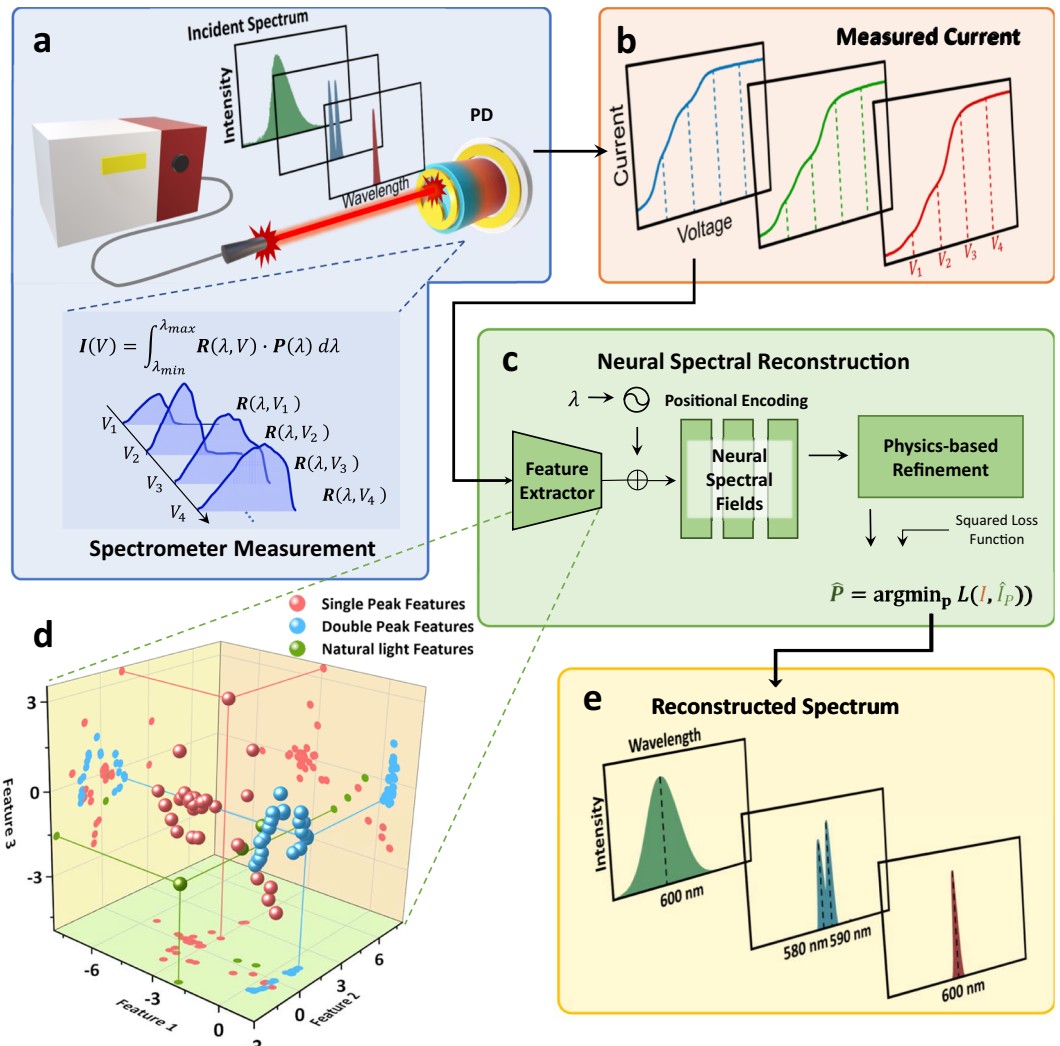

**Fig. 2 | Pipeline of the p-graded-n spectrometers. a** Schematic of photodetector absorption of incident light $P(\lambda)$ and generate photocurrent $I(V)$. Here, the incident light spectrum includes different types of single-peak, double-peak and natural light. **b** The measured current of three different types of incident light. Note that the current generated by different incident spectra shows different characteristics in shape features including turning points. **c** The Neural spectral reconstruction process flow. Our approach extracts the features in the measured currents and outputs the spectrum by neural spectral fields and physics-based refinements. **d** Visualization of the feature space. The three different types of incident light have unique distributions and are distinguishable from each other. **e** The reconstructed spectrum of three inputs with similar wavebands. Our spectrometer has the ability to reconstruct three different types of spectra with the tailored NSF method.

Fig. 3b. Our method achieves decent reconstruction performance for incident light with uncertain bandwidth, demonstrating its potential for a wide range of applications. It is noteworthy that while the absolute reconstruction error increases with a broader bandwidth, the relative error(normalized by bandwidth) remains consistent across different spectral widths. (More detailed reconstruction analysis can be found in Section S12 and S13 in Supplementary material).

The minimum double-peak resolution of the single-pixel p-graded-n junction spectrometer is about 10 nm, as demonstrated in Fig. 3c. To evaluate the resolving power of our method, we conduct a dense-sampling experiment on single-peak incident light with a spacing of 1 nm between adjacent spectra. The reconstruction result, shown in Fig. 3d, exhibits excellent performance in distinguishing peaks of spectra with narrow spacing. The reconstruction wavelength peak accuracy is approximately 0.30 nm. Our spectrometer also performs well in reconstructing more complex spectra, such as those with three or four peaks. For additional results, please refer to Supplementary S14.

Overall, these results demonstrate the remarkable potential of our single-pixel p-graded-n junction spectrometer for high-precision spectral reconstruction in a variety of settings.

## Spectral imaging results

We conduct experiments on an imaging target of a painting to explore the potential applications of our spectrometer in spectral imaging. As shown in Fig. 4a, a 180 μm diameter wire-bonded detector is placed on the image plane behind the lens and the device is swept along the X and Y axes using a motorized positioning system to cover the entire image area. (The footprint of our spectrometer can be found in Section S3 of Supplementary Material.) The imaging system is shown in Fig. 4b. The image plane is divided into multiple units. In each unit, the detector performs a current-voltage measurement. The X-Y sampling rate is set to be $100 \times 100$. The target image is a yellow Chinese character "Fu" on a red background, illuminated by a white LED light source. We reconstruct the image using the measured current-voltage data of the pixel, which is then mapped into RGB according to the CIE

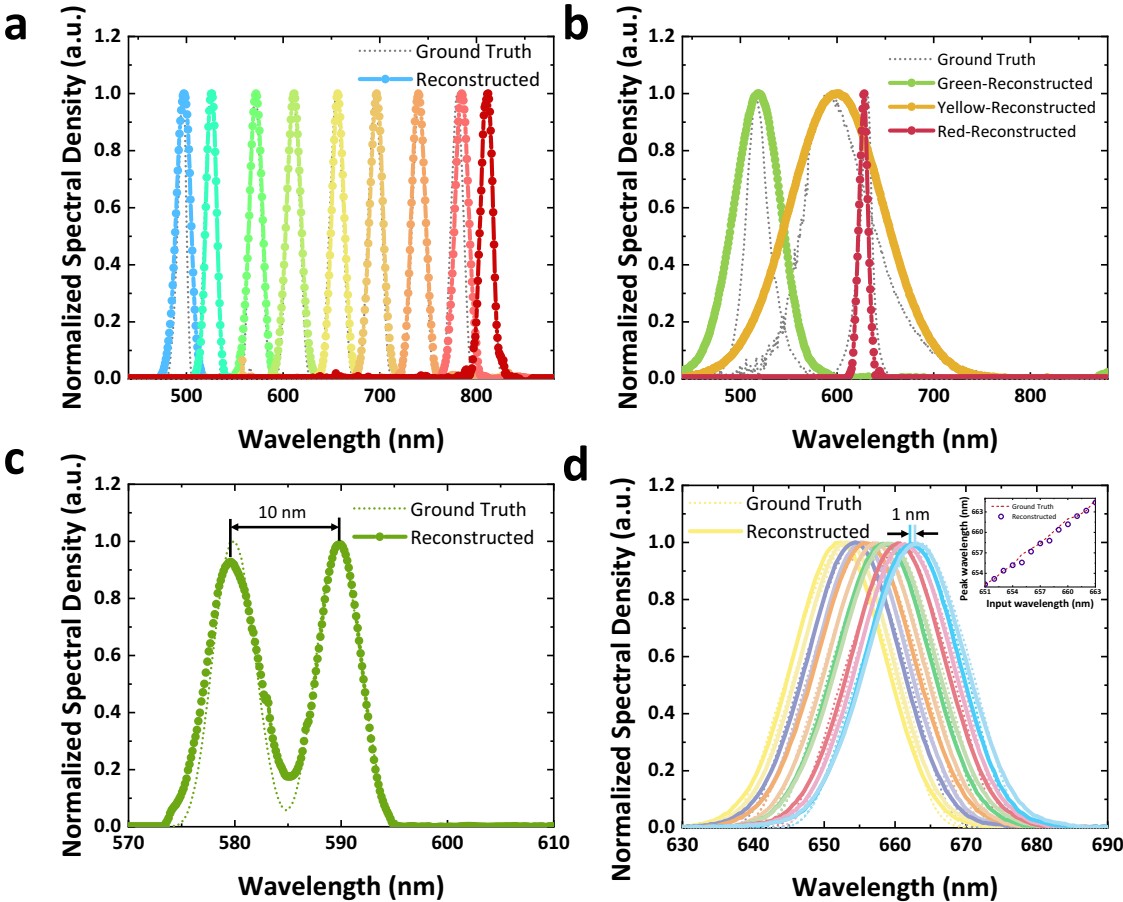

**Fig. 3 | Reconstruction results. a** Reconstruction results versus ground truth of single peak spectrum from 480 nm to 820 nm. **b** Reconstruction results of random bandwidth LED light source. **c** Double peak reconstruction result. **d** Dense-sampling measurement results. Sub figure on the right displays the peak accuracy in dense sampling reconstruction.

1931[35] color space mapping, as shown in Fig. 4d. Figure 4f is the spectral imaging results in different wavelength channels. The composited image is presented in Fig. 4e, compared with the image taken by CCD camera on the left. Our single-pixel p-graded-n junction spectrometer achieves high-quality spectral imaging within the functional waveband. The single-pixel spectrometer with this new scheme could easily be integrated on a chip with standard III-V process flow. Focal plane array (FPA) integrating mesa-structured pn junction device based on III-V semiconductor[29] are well-established for imaging. These FPAs comprise an inverted photodetector epilayer structure with substrate removed, coupled with a readout circuit, flip-chip bonded using indium bumps (Fig. 4h). Adapting our proposed p-graded-n junction into focal plane arrays enables the realization of hyperspectral imaging. This suggests the high feasibility of our solution in spectral imaging.

## Discussion

Miniaturization of optical spectrometers is crucial for various applications, including portable spectral detection, hyperspectral imaging, and wearable spectroscopy. In this study, we introduce a novel voltage-tunable-response III-V spectrometer based on the p-graded-n junction, which exhibits significantly improved device performance as a photodetector, particularly in terms of dark current and optical responsivity. We also propose a new method, Neural Spectral Fields (NSF), to reconstruct the spectrum from accumulated current, achieving high accuracy and robustness in spectrum reconstruction. Our work represents a major advancement in the miniaturization of optical spectrometers for diverse applications.

Future research will be aimed at realizing in-situ spectral imaging using arrays of single-pixel p-graded-n junction spectrometers with a readout circuit (ROIC). This 'spectrometer camera' provides multi-channel spectral information with an ultra-small footprint, without the use of a dispersive light path or detector array. This technology could push the boundaries of integrated hyperspectral imaging and find application in machine vision, aerospace detection and aerological sounding scenes.

## Methods

### Device fabrication

The epitaxial layer structure of the detector is grown with Metal-organic Chemical Vapor Deposition (MOVCD). Mesa structures of different diameters are fabricated by wet etching with $H_3PO_4$:$H_2O_2$:$H_2O$ solution (1:1:10). The detector diameter varies from 20 μm to 2 mm. Ti/Pt/Au contact metal is deposited with Denton EXPLOER-14 E-beam system on both P and N contact layers. A silicon dioxide ($SiO_2$) thin film is then deposited as the passivation layer. Detailed fabrication steps and flow can be found in Supplementary Material Section S5.

### Electrical characteristics measurement

**Dark & photo current.** The dark current and photo current is measured with Keysight B1500 semiconductor parameter analyzer at room temperature.

**Responsivity matrix.** The device responsivity matrix is measured by a halogen lamp passing through the monochromator (iHR 320,

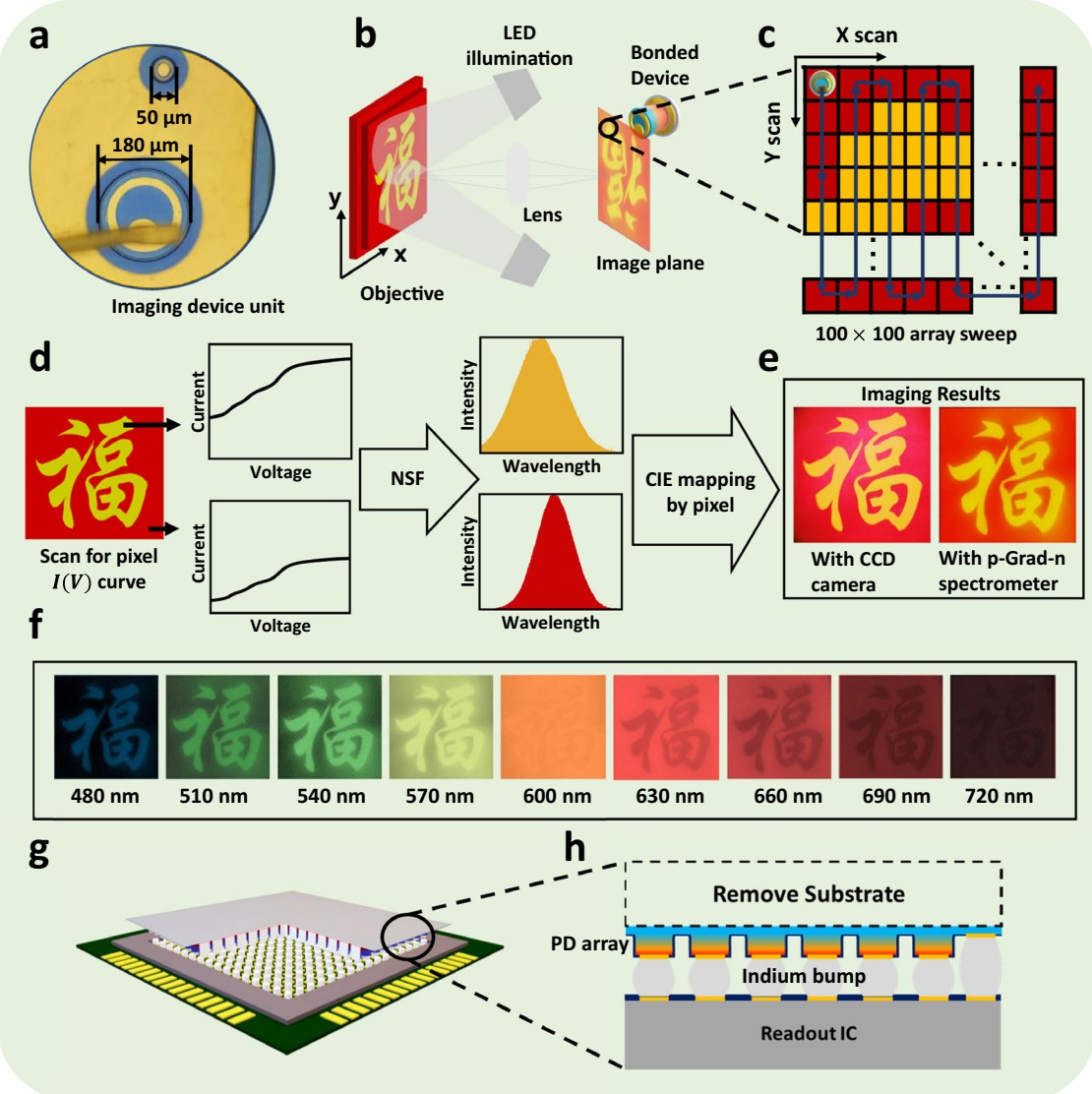

**Fig. 4 | Spectral imaging setup and results. a** Micro-graph of a wire-bonded single detector spectrometer of 180 μm diameter. **b** Imaging system schematic. The objective in the figure is a printed pattern on plain printer paper. **c** Scanning schematic of the device when imaging. **d** Data process flow of the spectral imaging. The current-voltage data is measured from different (x, y) positions. The reconstruction spectrum of each pixel was mapped into RGB color through CIE 1931 mapping. **e** Comparison of imaging results. Left: Image by CCD camera. Right: Reconstructed scanning spectral imaging result. **f** Spectral imaging results shown in different wavelength channels. **g** Vision of the integrated spectrometer array for hyperspectral imaging with a flip-chip bond focal plane array configuration. **h** Cross-section diagram of the focal plane array.

Horiba Instruments). The monochromatic light is modulated by a 180 Hz chopper and focused on the detector. The full width at half maximum (FWHM) of the monochromatic light for calibration is around 1 nm. The light spot size is about 50 μm. A source meter was applied to bias the detector and the chopped signal was measured by lock-in amplifier. Thorlabs FDS-1010 Si detector (with NIST traceable calibration) is used to calibrate the responsivity of the device.

**Current-Voltage curve for reconstruction.** During the measurement stage of the reconstruction experiments, the commercial LED light source of yellow, red, and green is applied to illuminate the detector. An NKT photonics supercontinuum laser is used to generate both single-peak and multiple-peak narrow-band light beams. We employ the commercial spectrometer of YOKOGAWA AQ6374 to measure the incident light spectrum as ground truth.

### NSF implementation

**Feature extractor.** We use a Multilayer Perceptron(MLP) to extract the features from measured currents. The MLP contains 12 hidden layers with 1024 neurons activated by the leaky rectified linear unit (leaky ReLU). Each hidden layer is attached with a batch-normalization layer and a skip connection to increase the capability of the MLP. The feature dimension is set to 501, which is enough and efficient with nanometer-resolution spectral reconstruction. The extracted features can be visualized to provide additional information. In Fig. 2, it is shown that the single-peak features and double-peak features are distinguished from each other. And broadband natural light features have a different distribution from single-peak features. Such properties in feature space provide NSF with the capability of reconstructing the spectrum of different spectra.

**Neural spectral fields.** We use the NSF to reconstruct the spectrum from the extracted features and given wavelength. The wavelength is

firstly positionally encoded to improve the network's ability to represent the high-frequency spectrum. The positional encoding process is expressed by:

$$\gamma_{pos}(\lambda) = \left[ \sin(2^0\pi\lambda), \cos(2^0\pi\lambda),..., \\ \sin(2^n\pi\lambda), \cos(2^n\pi\lambda) \right] \quad (2)$$

where $n$ is the dimension of positional encoding. We choose $n = 5$ in all experiments in our manuscripts. The positional encoded wavelength and extracted features are then fed into a small MLP with only 3 hidden layers and 256 neurons for each hidden layer. The output of the MLP is the spectral intensity at the input wavelength. Thus the NSF outputs the continuous spectrum giving the queried wavelength.

**Physics-based refinement.** The output spectrum from NSF is close to the actual spectrum, but it does not accurately correspond with the measured current. We use a physics-based refinement algorithm that iteratively adjusts the amplitude and scale to minimize the loss between the corresponding current and the measured currents. During each iteration, we randomly shift the NSF spectrum by 1 nm and scale the amplitude by 1.1 and 0.9. We repeat this process for 200 iterations for each input current. At the end of the refinement process, we select the spectrum with the lowest corresponding current error as the final result.

## Data availability
All data needed to evaluate the findings of this study are available within the Article. Source Data file has been deposited in Figshare under accession code 10.6084/m9.figshare.25194665[36]. Other data files related to the work and are available from the corresponding author upon request.

## Code availability
The code we used for reconstruction are provided with this paper in supporting information.

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

## Acknowledgements

The authors acknowledge the funding from the National Key Research and Development Program of China, 2019YFB2203400, and the National Natural Science Foundation of China under Grant 61975121. We are grateful to the device fabrication support from the ShanghaiTech University Quantum Device Lab (SQDL).

## Author contributions

B.C. conceived the idea. Jingyi Wang, B.P., Z.Z. and L.Y. carried out the preliminary verification of the idea. B.C and Jingyi Wang designed the device epi-layer structure. Jingyi Wang fabricated the devices, designed and participant in the experiment. B.P. developed the measurement code, participant in the measurement experiment. Z.W. developed the reconstruction code and algorithm pipeline. Y.W. and W.R. assisted Jingyi Wang and B.P. in spectral measurement. H.J. carried out the epi-layer growth. Jingyi Wang, B.P., Z.W., J.Z., Jianyu Wang, J.Y. and B.C. participant in the manuscript writing. B.C. and J.Y. supervised the research.

## Competing interests

The authors declare no competing interests.
