## [Peer Review File · Nature Communications]

Single-pixel p-graded-n junction spectrometersEditorial Note: This manuscript has been previously reviewed at another journal that is not operating a transparent peer review scheme. This document only contains reviewer comments and rebuttal letters for versions considered at Nature Communications.

Reviewer #1 (Remarks to the Author):

Summary of key results and comments to the manuscript

Optical spectrometry is one of the most powerful and widely used characterization tools in scientific and industrial research. Miniaturizing spectrometers down to submillimeter-scale footprint opens a range of opportunities for integration into lab-on-a-chip systems. Between the different strategies to reduce the size, computational spectrometer systems take advantage of more readily available computer processing power. In the manuscript, the authors demonstrate an ultracompact microspectrometer design based on a single compositionally engineered p-graded-n junction and computational spectral reconstruction.

The underlying principle in computational spectrometers is to integrate into a single nanostructure both wavelength responsivity and photodetection. In the manuscript, the spectral response function varies in terms of the reverse bias applied to a p-graded-n junction because of the gradual change in the composition of the semiconductor.

After that, the authors claim to solve the ill-posed inversion problem typical in computational spectrometers by application of the neural spectral field approach and subsequent physics-based refinement.

This work follows the path of previous designs, see for instance references 21 and 22 within the manuscript. As a main claim for the manuscript, it is strongly emphasized that the sensor is fabricated by standard III-V semiconductor material, in contrast to previously reported ultracompact spectrometers where "it remains challenging to grow large scale, high-quality two-dimensional materials or nano-wire for massive production". As a matter of fact, the junction is engineered such that the aluminum composition varies along their length showing a gradient bandgap and a spectral response that changes with reverse bias.

The performance of the system is tested by use of different light sources. Finally, a proof-of-concept experiment is carried out to demonstrate hyperspectral imaging by point scanning of the single-pixel sensor.

The main scientific message of the manuscript is clear, and the manuscript is easy to follow. Spectral measurement is at the heart of the manuscript and have been rationally designed to support the claims made in the paper.

After a first review, the authors have improved the details largely by including supplementary material. This additional information is strictly necessary for understanding the content of the manuscript so it should be mandatory it is included as an accompanying document (I couldn't find any reference to this supplementary information in the main manuscript).

The supplementary information fully addresses all the questions raised by the two reviewers. Main claim of the manuscript is put in context and compared with existing solutions, so that one clearly appreciates the strengths of the contribution which, indeed, represent a step forward in the field (superior responsivity and lower dark current).

The authors provide fabrication details and the underlying physical mechanism. Scalability and device-to-device reproductivity are addressed satisfactorily. Also, the description of the algorithms and the

training procedure are convincing.

Some experiments have been performed with success to test operational behaviour (noise sensitivity, linearity, spectrum complexity, ...).

I consider the manuscript adequate for Nature Communications. However, I feel that the authors should do an additional effort to address the above points:

1. Concerning measurement of spectrum. In fact, it is not clear to me the response to question 7 raised by reviewer#2. Figure 9 in the rebuttal letter is extremely perturbing, as negative wavelengths are mentioned and included in the plots. Aside from this detail, a stronger explanation about the reconstruction accuracy regarding the width of the peak should be provided. Where does the reconstruction error come from?
2. Concerning spectrum complexity. Figure 10B shows that reconstructed results are worse for a four-peak spectrum. Again, a further analysis should be provided. Is it related with the synthetic dataset used for training?
3. Concerning spectral imaging, the reported experiment is still far away from a practical application (eight hours for a 100x100 scanning). In the rebuttal letter, the authors claim that the long acquisition time is due to the "long-motorized stages movement time" in their experiment. Are the 3 seconds for a point measurement competitive regarding existing technologies?

Minor considerations:

In the first reference of the supplementary material, there is some information missing:

1] Adachi, S.: GaAs and Related Materials. WORLD SCIENTIFIC, ??? (1994). <https://doi.org/10.1142/2508> . <https://www.worldscientific.com/doi/abs/10.1142/2508>

Reviewer #2 (Remarks to the Author):

The authors have addressed my comments with plenty of new content. This manuscript is much improved now.

Point-by-point response

Reviewer #1 (Comments to the Author):

Reviewer comment:

After a first review, the authors have improved the details largely by including supplementary material. This additional information is strictly necessary for understanding the content of the manuscript so it should be mandatory it is included as an accompanying document (I couldn't find any reference to this supplementary information in the main manuscript)

Our Response:

Thank you very much for your comments. The reference directions to the supplementary information section are marked red in the main manuscript, such as:

“Please refer to Supplementary S2 for bandgap of $\text{Al}_x\text{Ga}_{1-x}\text{As}$ material.”

“Please refer to Supplementary S1 for more details of the working mechanism and advantages of the p-graded-n junction spectrometer.”

“Please refer to Supplementary S9 for details of response function characterization.”

Reviewer comment:

1. Concerning measurement of spectrum. In fact, it is not clear to me the response to question 7 raised by reviewer#2. Figure 9 in the rebuttal letter is extremely perturbing, as negative wavelengths are mentioned and included in the plots. Aside from this detail, a stronger explanation about the reconstruction accuracy regarding the width of the peak should be provided. Where does the reconstruction error come from?

Our Response:

We thank you for your professional comments and advice. For reference, the question 7 raised by reviewer#2 is:

7. In Fig. 4b, the experimental results indicate that the reconstruction accuracy gets worse with broader peaks. Please explain that.

Fig.4b

	Red	Green	Yellow
Spectrum width	16 nm	60 nm	108 nm
Reconstructed peak error	2.6 nm	5.3 nm	12 nm
Reconstructed width error	2 nm	29 nm	34 nm

Table 2 in the rebuttal letter

We apologize for any confusion caused by Figure 9 in our rebuttal letter. We have revised Table 2 to better illustrate that the reconstructed error increase as the spectrum width widens. We believe this is due to the growing discrepancy between real and synthetic data as the spectrum width expands. Given that the neural network is trained on synthetic data, its performance degrades as the spectrum width enlarges. We additionally demonstrate our opinion on experimental data with different spectrum width.

Figure S9(b) shows that as the spectrum width broadens, the measured I-V curve does not perfectly represent the ideal I-V curve, defined as the product of the response function and the spectrum measured by commercial spectrometer. This discrepancy may stem from the inherent response function of the ground truth from the built-in detector in the commercial spectrometer. With a small spectrum width, this inherent response function is almost a delta function and has minimal impact. However, as the spectrum width expands, the influence of this inherent response function grows. Consequently, the synthetic training data becomes biased and lacks the exact distribution of real experimental data as the spectrum width increases, leading to a slight decline in neural network performance.

Figure S9 in the revised supplementary information

We also revised the supplementary material with a section **Broadband reconstruction** to fully address the question above. Please kindly refer to **Broadband reconstruction** for this part of revision as below.

“We have examined the reconstruction error, as depicted in the Fig. S9. Our evaluation of the spectrum reconstruction performance in relation to the peak width reveals that the reconstruction error increases as the peak width expands. Fig. S9 (a) and (b) show the reconstructed I-V curve derived from algorithmic prediction, while the ideal I-V curve is the product of the response function and the ground-truth spectrum. It should be noted that the ideal I-V curve can be considered as reference.

The accuracy of the reconstruction in terms of peak width should be considered from two perspectives: unreal synthetic data and algorithm reconstruction error. Firstly, the measured I-V curve doesn't perfectly represent the product of the response function and the spectrum. This discrepancy could be due to the inherent response function of the ground truth from built-in detector in the commercial spectrometer. Consequently, the synthetic data for training is biased and lacks an exact distribution from the real experimental data. As illustrated in Fig. S9 (a) and

(b), the real I-V curve diverges from the ideal one, and the reconstruction error escalates with the increase of difference between real I-V curve and reconstructed one.

Second, Fig. S9 (c) exhibits slight distortion as the peak width increases, which can be attributed to the laser source used in the experiments. Given that the single peak spectrum is primarily modeled as a Gaussian function in synthetic data construction (please refer to the "Synthetic dataset construction" section in the supplementary material for details), it is reasonable that the reconstructed I-V curve deviates from the actual one. Fig. S9 (d) provides a more detailed analysis of the relationship between different errors. The dataset error is defined as the Mean Average Error (MAE) between the real and ideal I-V curve. The spectrum error is defined as the MAE between the reconstructed spectrum and the ground truth. The algorithm error is defined as the MAE between the reconstructed and real I-V curve. It is evident that the spectrum error increases as both the dataset error and algorithm error increase."

Reviewer comment:

2. Concerning spectrum complexity. Figure 10B shows that reconstructed results are worse for a four-peak spectrum. Again, a further analysis should be provided. Is it related with the synthetic dataset used for training?

Our Response:

We appreciate your advice and prospective opinion. We attribute the worse results of the four-peak spectrum to two factors: the complexity of the four-peak spectrum, and the discrepancy between synthetic and real experimental data.

Revised Figure 10(a) and (c) shows that the reconstructed three-peak spectrum is well-executed, with the corresponding I-V curve aligns well with the real one.

The four-peak spectrum is significantly more complex than the three-peak spectrum. Consequently, Figure 10(b) shows that our method only reconstructs the main peak of the four-peak spectrum, and is not as precise as three-peak spectrum.

Additionally, the discrepancy between synthetic data and experimental data worsens the results. Figure 10(d) reveals that the I-V curve, reconstructed using our algorithm, does not align well with the real I-V curve. This is because that the neural network is trained on synthetic data and tend to reconstruct the ideal I-V curve over the real one, leading to a less accurate reconstructed four-peak spectrum.

Figure S10 Results and analysis of multi-peaks.

We also added a section **Multi-peak reconstructions** in the supplementary material for this part of discussions. Please kindly refer to the section **Multi-peak reconstructions** in supplementary materials as shown below.

“In order to further assess the effectiveness of our method on more complicated spectra, we employed our device to detect incident light with three and four peaks. We first train the neural network using only three-peak and four-peak synthetic data respectively.

Fig.S10(a) and (c) display the reconstructed spectrum and I-V curve of three-peak reconstruction tasks. Fig.S10 (a) reveals that our neural network method effectively reconstructs the three-peak spectrum, albeit with some existing artifacts. Fig.S10 (c) further demonstrates that the reconstructed I-V curve aligns closely with the real I-V curve, suggesting that the algorithm has accurately learned the physical constraints of the spectrometer model. However, the real I-V curve deviates from the ideal one, indicating that the synthetic dataset used for training differs from the real ground-truth experimental data, leading to artifacts and errors in the reconstructed spectrum as shown in fig.S10 (a). The discrepancy between the real and ideal I-V curve could arise from the intrinsic response function of the commercial spectrometer used to measure the ground truth spectrum.

Fig.S10 (b) and (d) present the reconstructed spectrum and I-V curve of four-peak reconstruction tasks. Fig.S10 (b) shows that our neural network method can reconstruct the main peak of the four-peak spectrum. However, the reconstruction of the four-peak spectrum is

not as precise as that of the three-peak spectrum. Fig.S10 (d) provides an explanation for this. It shows that the reconstructed I-V curve derived from the algorithm does not fit the real I-V curve well. This could be attributed to the higher complexity of the four-peak spectrum compared to the three-peak spectrum. Given that the proposed physics-based refinement only applies simple operations such as shift and scale, the reconstructed spectrum, with its artifacts, does not perfectly fit the real I-V curve.”

Reviewer comment:

3. Concerning spectral imaging, the reported experiment is still far away from a practical application (eight hours for a 100x100 scanning). In the rebuttal letter, the authors claim that the long acquisition time is due to the “long-motorized stages movement time” in their experiment. Are the 3 seconds for a point measurement competitive regarding existing technologies?

Our Response:

Thank you very much for your valuable comments. Our intention in designing this experiment was to demonstrate the potential of the P-graded-N junction spectrometer for hyperspectral imaging applications. It is noted that the imaging time in this specific experiment may not serve as an accurate reference, as the future developed imaging system based on the P-graded-N junction focal plane array (FPA) is anticipated to have significantly faster imaging speed.

Regarding a single imaging process, the 3-second measurement in this work was caused by measurement setup. The current measurement of this work was conducted using a Keithley 2400 source meter. The computer sends bias value commands to the source meter and reads the corresponding current values via GPIB connection under IEEE-488 principles, following a handshake communication mode[1]. It's important to note that the measurement command sending and processing introduce time delays for every input-measure-output operation in the measurement process. Therefore, **this 3-second measurement time for each point measurement includes motorized stage movement time, GPIB communication time for each bias, Keithley measurement time.**

Since the FPA imaging array with a Digital-to-Analog Converter (DAC) can operate at an update rate of up to 1 megasample per second, and transfer the measured current using an Analog-to-Digital Converter (ADC) with a similar rate, it is expected have an imaging time with a P-graded-N junction photodetector array of approximately 10 microseconds (μs). That would be competitive to the existing technologies.

Reviewer comment:

Minor consideration:

In the first reference of the supplementary material, there is some information missing:

1] Adachi, S.: GaAs and Related Materials. WORLD SCIENTIFIC, ??? (1994).

<https://doi.org/10.1142/>

2508 . <https://www.worldscientific.com/doi/abs/10.1142/2508>

Our Response:

Thank you for your comments. We have corrected the citation information that you mentioned in the supplementary material:

[1] Adachi, S.: GaAs and Related Materials, 2nd edn. WORLD SCIENTIFIC, P O Box, 128 Farrer Road, Singapore, 912805 (1994). <https://doi.org/10.1142/2508>.

Please kindly refer to **References** in Supplementary Materials.

Reviewer #2 (Comments to the Author):

Reviewer comment:

1. The authors have addressed my comments with plenty of new content. This manuscript is much improved now.

Our Response:

We sincerely appreciate your valuable comments and suggestions, which have greatly improved our manuscript.

References

[1] “Basic Knowledge and Glossary for GPIB Communication,” CONTEC. Accessed: Jan. 19, 2024. [Online]. Available: <https://www.contec.com/support/basic-knowledge/daq-control/gpib-communication>

Reviewer #1 (Remarks to the Author):

The authors have addressed my comments satisfactorily. The revised manuscript is of high-quality and represents important advances in the field of spectrometry. In my opinion, the manuscript fits the requirements of Nature Communications.